# Integrated multilayer stretchable printed circuit boards paving the way for deformable active matrix

Shantonu Biswas[1], Andreas Schoeberl[2], Yufei Hao[3], Johannes Reiprich[2], Thomas Stauden[2], Joerg Pezoldt[2] & Heiko O. Jacobs[2]*

Conventional rigid electronic systems use a number of metallization layers to route all necessary connections to and from isolated surface mount devices using well-established printed circuit board technology. In contrast, present solutions to prepare stretchable electronic systems are typically confined to a single stretchable metallization layer. Crossovers and vertical interconnect accesses remain challenging; consequently, no reliable stretchable printed circuit board (SPCB) method has established. This article reports an industry compatible SPCB manufacturing method that enables multilayer crossovers and vertical interconnect accesses to interconnect isolated devices within an elastomeric matrix. As a demonstration, a stretchable (260%) active matrix with integrated electronic and optoelectronic surface mount devices is shown that can deform reversibly into various 3D shapes including hemispherical, conical or pyramid.

[1] California NanoSystems Institute, Elings Hall, Building 266, Mesa Road, University of California, Santa Barbara, CA 93106-6105, USA. [2] Fachgebiet Nanotechnologie, Institut für Mikro- und Nanotechnologien MacroNano®, Technische Universität Ilmenau, Gustav-Kirchhoff-Strasse 1, D-98693 Ilmenau, Germany. [3] School of Mechanical Engineering and Automation, Haidian District, Beijing Institute of Road 37, Beihang University, 100191 Beijing, China. *email: heiko.jacobs@tu-ilmenau.de

A dramatic increase in research activities has been perceived for last decade to enable mechanically stretchable and deformable functional electronic devices[1–4]. Consequently, a large number of stretchable devices have been realized demonstrating a wide range of diverse applications that includes soft robotics[5,6], actuators[7], electronic eye cameras[8], epidermal electronics[9], wearable electronics[10,11], metamorphic electronics[12,13], edible electronics[14], acoustoelectronics[15,16], health monitoring devices[17–19], smart textiles[20] to give a few examples. Most of these demonstrators often use highly specialized technologies and unconventional materials that make these technologies more interesting for research, but less favorable for industrial production for mass people. Till today, there exist no reliable manufacturing methods that can be generalized as stretchable printed circuit board (SPCB) technology[21].

Conventional rigid printed circuit boards (PCB) typically consists of more than one metallization layers to route metal tracks to interconnect surface mount devices (SMDs) using well-established manufacturing methods, which is one of the main reasons behind the paramount success of this technology. On the other hand, stretchable electronics mostly remains limited to a single active layer with less complex device integration, which is primarily due to the lack of reliable manufacturing methods. Although, there are a few lab prototypes of stretchable devices demonstrating multilayer electronic systems with different functionalities[22–24], the materials and methods used to realize such devices are unconventional and rarely suitable for industrial production. This technological and materials lacking confines the complexity of demonstrated stretchable electronic devices[25]. For instance, even the simplest functional active matrix requires at least two metallization layers[26].

Additionally, vertical interconnect accesses (VIAs) are required to interconnect between different active layers in the circuit boards, which is not well-established in the manufacturing process of stretchable electronics. Although, a few approaches have been reported to realize VIAs in stretchable substrates using liquid alloys[27,28] or solid phase materials[22,29], again, the methods and the materials are incompatible to conventional processing. Thus, an industry-compatible processing of multilayer metal tracks and reliable VIAs, which are two important elements to realize a SPCB technology, remains a primary challenge.

Recently, we demonstrated a single layer SPCB method that enabled "on-hard-carrier" fabrication using conventional planar microfabrication techniques that delays use of elastomeric substrate to the end[30]. The reported method enabled high temperature processing, high alignment and registration, and allowed conventional chip assembly methods on a rigid carrier. However, the previously demonstrated methods used only a single active layer without complex routing of the metal tracks, which limited the complexity of the circuit and the device[12,13,30]. In this article, we engineered a similar method to realize integrated multilayer SPCB and demonstrate an alternative development towards the realization of stretchable electronics with higher integration density capabilities by introducing stable VIAs through interconnection between different metallization layers. The method used in this article is compatible with conventional micro fabrication processes and uses commercially available pristine SMDs.

Here, we report a SPCB method which replaces the rigid insulator substrate of conventional PCB with a highly stretchable silicone elastomer (EcoFlex). Demonstrated manufacturing method to realize SPCB can be divided in two steps, the first step uses hard-carrier fabrication method that is fully compatible with conventional planar microfabrication techniques, where the second step introduces elastomeric substrate and no active fabrication is required during this. This method has several benefits over other demonstrated methods in the field of stretchable electronics[22] since it delays the introduction of rubber substrate which enables high temperature processing, higher registration and alignment accuracy, and allows to use conventional robotic or advanced self-assembly of SMD dies[31]. Additionally, the method allows testing the device functionality on hard-carrier which is beneficial since it allows for the identification of failure modes of the circuit and device layer before and after we detach, bend or stretch the structure. Moreover, like conventional PCB technology, this method allows direct use of SMD chips. To realize VIAs in the SPCB, a similar method to the conventional PCB technology is used here. A highly stretchable (elongated up to 260% of the original length) multilayered integrated SPCB design is discussed. To demonstrate the applicability, a fully addressable LED active matrix has been realized. The integrated LED display can be deformed to various three-dimensional (3D) geometrical shapes to morph hemisphere, cone, and pyramid.

## Results

**On hard carrier fabrication.** Figure 1 shows design and first part of the multilayer SPCB manufacturing method on a hard carrier. As an example, an active matrix LED display segment is realized. As mentioned, active matrix LED array requires at least two metallization layers, VIAs, and the integration of transistors and LEDs (1a) in an array type fashion. From materials and processing point of view, several elements are important to achieve this. Figure 1b schematically presents elements of the first step of SPCB manufacturing process on hard-carrier. The depicted method uses a rigid Si wafer (500 μm thick, MicroChemicals, Ulm, Germany) as a carrier substrate and forgoing processing are performed on this rigid substrate. The details of the processing are added in the methods and supplementary methods (Supplementary Fig. 1).

*Release and peeling layer*: Even though there are several benefits of using a rigid carrier substrate for the processing, one drawback of this method is that the final device has to be detached from the rigid substrate to be stretchable. A few other demonstrators use sequential transfer methods to retrieve different parts of the device such as metal tracks, active components etc. from rigid carriers to a stretchable substrate. This approach becomes highly challenging when active elements of the device become fairly small due to the alignments and registrations, and thus achieving higher integration density becomes difficult. In the depicted approach, we realize the complete functional electronics on-hard-carrier and detach the entire device to a stretchable substrate by a single step transfer technique using predefined two sacrificial layers of poly(methyl-methacrylate) (PMMA) and polyimide (PI). In this approach spin coated PMMA, (1 μm thick, green in Fig. 1b) and an eight μm thick layer of PI (blue in Fig. 1b) is used as a release and a peeling layer, respectively. The peeling layer supports the buildup of the circuit and enables detachment of the circuit after the fabrication is completed.

*Metal 1*: One of the major elements of the stretchable electronic devices is the conductive interconnects where rigid SMDs are used, since the interconnects directly contribute to the stretchability of the device. In the current demonstrator, we used 10 μm thick copper (Cu) tracks patterned as stress-adaptive meander shaped[32]. Initially a 50 nm/200 nm thick sputter coated seed layer of Al/Cu is deposited, which is patterned by photolithography to electroplate 10 μm thick layer of Cu (Supplementary Fig. 2) to increase the mechanical robustness of the metal tracks (reddish in Fig. 1b). Thick metal tracks (>5 μm) were found to be more robust than previously used thin (<1 μm) metallization layers[33]. This metallization layer forms the columns in the addressing system (Fig. 1c). Moreover, a part of it (50 nm Al) serves as a self-aligned etch mask in a later plasma etching step, necessary to

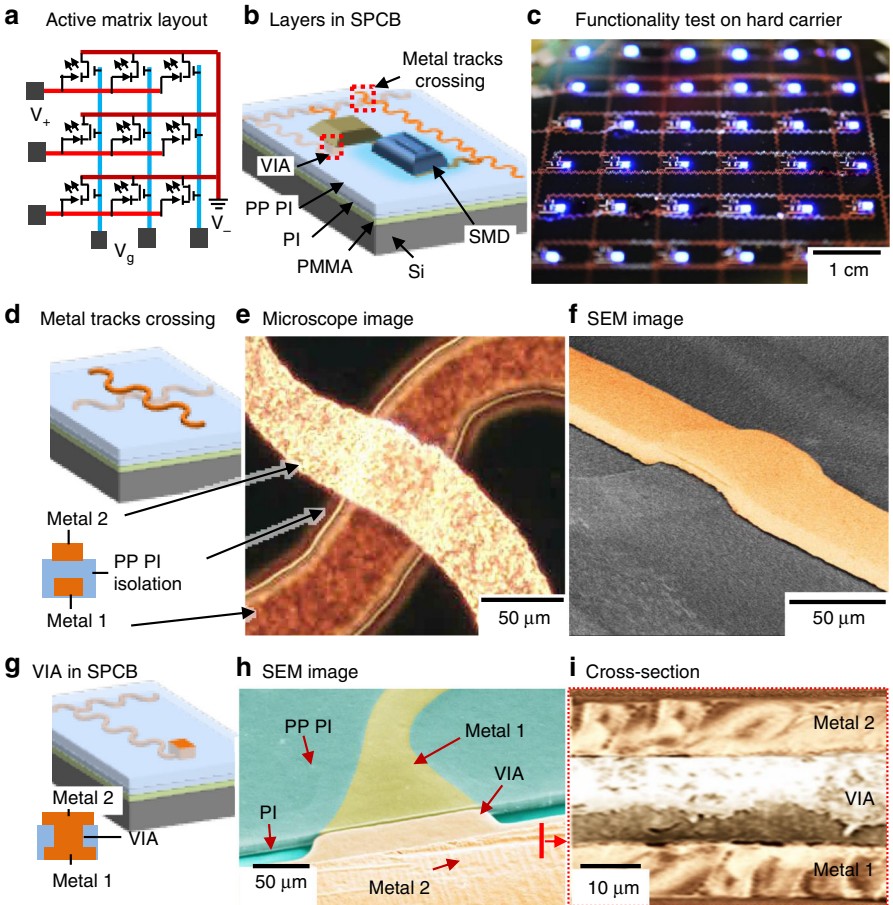

**Fig. 1** First steps to realize integrated stretchable printed circuit boards. Schematic layout of an active matrix using LEDs and transistors (**a**), layers build-up in the SPCB (**b**) and "on-hard-carrier" functionality test (**c**). Metal layers crossing: schematics (**d**) next to corresponding optical microscope image (**e**), and colored SEM image (**f**); the two metal layers are separated by a 20 μm thick photo-patternable PI layer, the layer is black in the microscope and gray in the SEM image. VIA: schematics (**g**) next to a colored tilted SEM image (**h**) depicting the PI peeling layer (turquoise), bottom metal (yellow) (in parts covered by photo-patternable PI, (darker turquoise), VIA (reddish), and top metal (reddish) and cross-section (**i**) of the same. *PI* polyimide, *PMMA* poly(methyl-methacrylate), *PP PI* photo-patternable PI, *SMD* surface mount device, *VIA* vertical interconnect access, *SPCB* stretchable printed circuit board

remove the sacrificial PI layer and the photo-patternable PI insulation layer.

*Insulation layer*: Multilayer electronic systems require electrical isolation in between different metal tracks to prevent electrical short-circuits. In the depicted case, the rows and the columns are separated using a photo-patternable PI layer (HD 4100) layer. Specifically, a spin-coated 20 μm thick photo-patternable PI layer is used to entirely cover the first metal tracks. As illustrated in the figure, the intermediate photo-patternable PI layer (Fig. 1d) serves as an insulation layer. The optical microscope image (Fig. 1e, Supplementary Fig. 3) shows a top view of one of the crossing regions. The bottom metal appears darker, because it is buried underneath the photo-patternable PI layer. The colored SEM image (Fig. 1f) shows a tilted view. The photo-patternable PI layer follows the surface profile and covers the top and sidewalls of the first metal tracks. This results in a raised surface, and this surface topography continues to the second metallization layer, which produces the visible crossing. The thickness of the insolation layer is 20 μm. This layer is selectively removed by plasma etching after encapsulation and detachment using the first metal tracks (Al) as a mask. As a result, in the final device the isolation layer has a similar geometry as metal 1 and can be stretched. Electrically we found no short-circuits or leakage currents between individual rows and columns before or after the detachment of the device.

*VIAs*: As mentioned earlier, metal tracks in different layers require to be interconnected through VIAs and realizing reliable VIAs in SPCB remained a challenge till today. In the depicted approach, like the conventional PCB method, we use electro-plated Cu to realize the VIAs. The photo-patternable PI layer is also photolithographically patterned to define openings and locations of the VIAs, which are subsequently filled with 20 μm of electrodeposited copper. This second electrodeposition step is necessary in order to ensure good electrical contact in between the first metal tracks and the VIAs, which is difficult to achieve using thin-film deposition methods through 20 μm deep holes. A similar electrodeposition method is also used in conventional PCB technologies to grow thick VIAs. As shown schematically in Fig. 1g, the VIAs are grown using metal 1 as a seed layer through predefined openings in the photo-patternable PI layer. A plasma cleaning process is required prior to electrodeposit the VIAs in order to remove the residues from photo-patternable PI (Supplementary Fig. 4) to ensure good electrical and stable mechanical contact between different metal layers in these regions.

*Metal 2*: Second metal tracks are directly deposited on top of the isolation layer and the VIAs ensuring good electrical contact between metal 1, VIA, and metal 2. Another 10 μm thick layer of Cu is used as the second metallization layer (reddish layer). The colored SEM image (Fig. 1h) shows a tilted view (and optical

microscope image in Supplementary Fig. 5 shows top view) of the VIA region, revealing the PI peeling layer, metal 1, the photo-patternable PI layer, VIA, and metal 2, and the cross-sectional SEM image (Fig. 1i) shows different metal layers in the VIA region.

*Solder bump application*: The solder bumps are used to make mechanical and electrical contact in between the metal tracks and the active components. A low melting point solder (Indalloy #117) is used for this purpose because the solder is applied by a parallel dip-coating process in a liquid solder bath[34] using a 10 µm thick photoresist mask to define the solder bump locations (see Supplementary Fig. 6). However, a higher melting point solder could be used as well following other solder printing methods.

*Component assembly*: The fabrication process is compatible with various types of chip attachment and component assembly methods, including solder bump-based interconnects[35], flip-chip die attachment, robotic pick-and-place or engineered self-assembly using molten solder[31]. In the demonstrated case, we use a semi-automated pick-and-place process to mount the components. This demonstrator contains a number of lab-fabricated (bare dies, requiring flip-chip die assembly) field effect µ-transistors (µ-FET, see Supplementary Fig. 7) ($0.5 \times 0.5 \times 0.5$ mm) and an equal number of commercially-bought standard surface mount LEDs ($1 \times 0.6 \times 0.2$ mm, 459 nm, Creative LED GMBH, Schaan, Fürstentum Liechtenstein).

*On hard carrier functionality tests*: The depicted approach enables "on-hard-carrier" functionality tests. This is different from other methods, which build on a soft elastomeric rubber substrate. This is beneficial since it allows for the identification of failure modes of the circuit and device layers before and after we detach, bend, or stretch the structure. For example, in the depicted hard carrier functionality tests (Fig. 1c) all the LEDs function properly and response to the addressing system.

**Introducing elastomeric substrate**. In this step, an elastomeric material is introduced in liquid form which will become the final stretchable substrate after curing. A wide range of elastomeric resin is available including silicone or plastic which make this method compatible with a large variety of substrate materials with different mechanical and optical properties. Mainly two processing steps are involved in this part.

Encapsulation and detachment: To detach the device layer from the hard carrier, we use a castable 3 mm thick and thermo-curable (room temperature, 15 h) layer of EcoFlex (Smooth-On, EcoFlex 00-30) as a stretchable encapsulation layer which is poured evenly over the entire surface of the fabricated device. To increase the bond strength between the active layers and the EcoFlex, a preceding 5 min long $O_2$ plasma activation step is used. The process is carried out without any mask, meaning the whole surface is exposed to the plasma which includes the isolation layer, metal 2, solder, and the SMDs. In general, the plasma cleans the surfaces, increases the surface roughness and activates the polymer surface with increasing –OH group.

As shown schematically in Fig. 2a (and Supplementary Fig. 8), the molding process encapsulates the SMDs, meaning the EcoFlex under fill the SMDs. It is known that elastomers with high viscosities will require high local pressures to fill in small gaps. However, we observed that EcoFlex 00-30 (viscosity, $\eta = 3000$ cP) can fill smaller gaps than 10 µm (see Supplementary Fig. 9). Comparing with Polydimethylsiloxane (PDMS), which is well known for micro-patterning through soft-lithography, has a viscosity of 3500 cP. Both of these two polymers are strong candidates as a stretchable substrate for high dense stretchable electronics.

The detachment process uses the differential interfacial adhesion of the stacked layers through an interface that can be detached. Specifically, the PI layer has a low level of adhesion to the PMMA coated carrier and this interface (PMMA ↔ PI) detached during this step. The detachment process works particularly well using the introduced PI film, which forms a uniform non-stretchable and supporting peeling layer beneath the circuit. Figure 2b shows an image of the detachment process while the LEDs are turn on and no damage to the device is introduced during this process. It should be noted that no solvent is required during the detachment process which increases the robustness of the method since some electronic components get damaged in some solvents.

Removal of unwanted PI: After detachment, the Cu metal tracks continue to be covered with the PI peeling foil (schematics in Fig. 2b inset and Supplementary Fig. 10), which needs to be removed. Moreover, large sections of the intermediate PI foil are treated as an unwanted layer, because a continuous film of PI reduces the stretchability of the system. We used electron cyclotron resonance (ECR) plasma etching process (40 SCCM $O_2$, 10 SCCM $CF_4$, 100 W RF power, 30 min at 0.025 mbar) to accomplish the removal of these two layers. The interesting part is the second intermediate photo-patternable PI layer which is used as an isolation layer between two metal layers. This layer will be removed everywhere, except for the region that is covered with metal 1. The first metal track acts as hard mask during the plasma etching process which means that the photo-patternable PI layer will have the same stretchable-meander shape structure as Metal 1. The image in Fig. 2c shows that all the LEDs are lighting in the EcoFlex substrate after completing the entire fabrication process.

**Device in rubber matrix**. Figure 2d–f shows images of a single pixel, crossing of two metal tracks, and VIA, respectively, in the SPCB in EcoFlex matrix. The active devices remain completely embedded in EcoFlex (surrounded and under-filled), which provides protection during final stretching operations. The close-up image reveals the bottom side of a single pixel in the active matrix showing the transistor and the LED (Fig. 2d). The bottom provides access to the metallization layer, meaning the structure has a surface mount like geometry. Pads are accessible from the bottom and all other elements are embedded and surrounded with silicone; this is different from methods that build on top of an elastomeric carrier. The SEM image (Fig. 2e) shows the crossing of two metal tracks in the EcoFlex. After detachment, the metal 1 becomes on top and metal 2 encapsulates in the EcoFlex matrix. The insulator (photo-patternable PI layer follows the shape of the meander of metal 1 and no critical alignment is required to prevent short-circuits. Figure 2f shows a SEM image of a VIA intentionally lifted off from the EcoFlex substrate connecting two metal tracks.

**Reliability of VIAs**. The VIAs play a major role in the system with more than one metallization layers, and are the key challenges in the field of multilayer stretchable electronics. Specifically, the area of the VIAs had to be optimized in order to achieve fully functioning arrays. The results of this optimization are summarized in Fig. 3 with computer simulated stress profile at the VIA locations while stretched. VIAs connecting bottom and top metal track in an open location (Fig. 3a, b) and VIAs connecting a metal track to one of the contact pads of a component (Fig. 3c, d) can be distinguished. A goal was to establish the maximum level of uniaxial elongation of the system to cause an electrical discontinuity. The measured values of the elongation ranged up to 260% of the original length can be

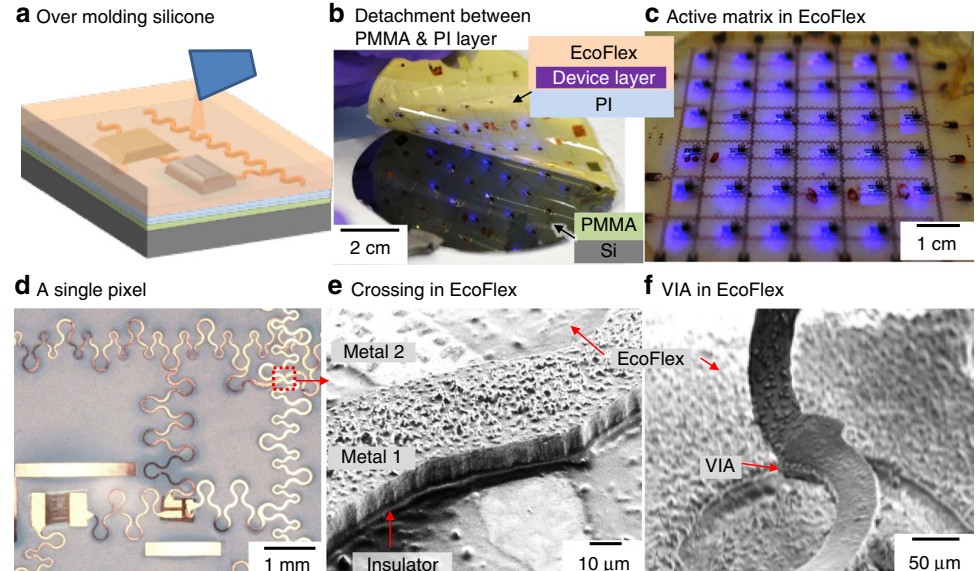

**Fig. 2** Second steps to realize integrated stretchable printed circuit boards. Schematic of silicone encapsulation of the device on hard carrier **a**. A photograph of the detachment process of the device layer from the hard carrier while the device is under operation (**b**) and functionality test of the final device in EcoFlex (**c**). Device in rubber matrix: Photograph of a single pixel of the active matrix encapsulated in EcoFlex showing a transistor and a LED (**d**). SEM image of a crossing of two metal layers in EcoFlex (**e**) and a VIA lifted off from the silicone substrate connecting two metal tracks (**f**)

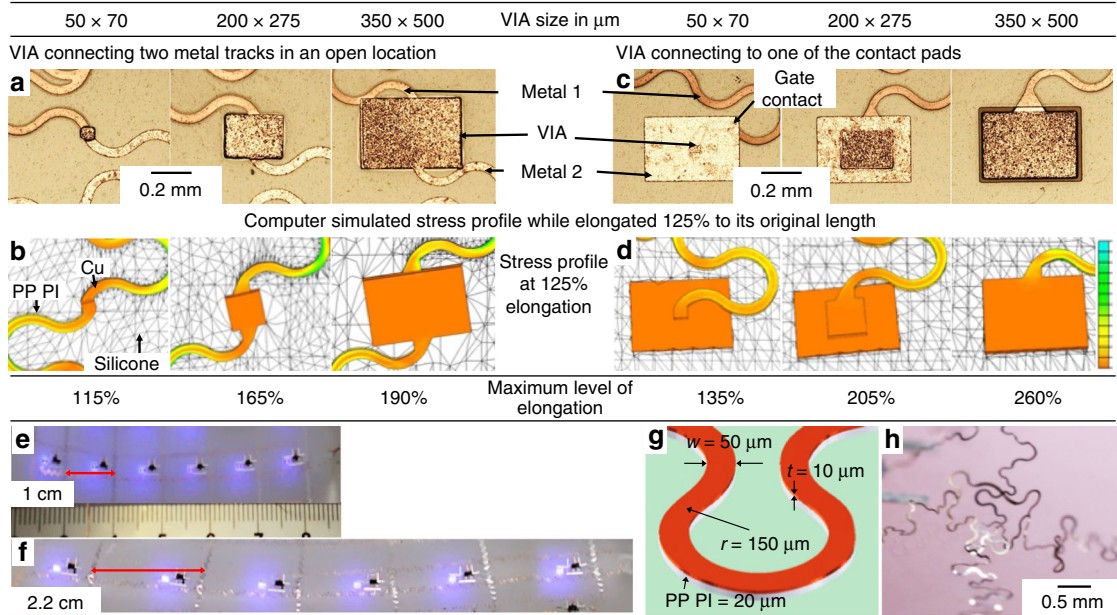

**Fig. 3** VIA designs affecting the maximum level of elongation. **a**, **b** VIAs connecting bottom and top metal tracks in an open location and **c**, **d** VIAs connecting a metal track to one of the contact pads of a SMD. Optical microscope photographs of the VIAs (**a**, **c**) and computer simulated stress profile in the metal tracks while elongated 125% to its original length (**b**, **d**). The dimensions of the VIAs and corresponding maximum level of uniaxial elongation are shown in the table. **e**, **f** Results of the stretching tests using a stress adaptive meander shaped metal track in unstretched condition (**e**) and while elongated to 220% (up to 260%) to its original length (**f**). **g** Schematic dimensions of the meander shaped metal tracks and **h** image of a failure mode in the SPCB using current design. VIA vertical interconnect access, PP PI photo-patternable PI

achieved. Considering VIAs to the contact pads, it was found that maximizing the footprint is beneficial. An increase in the VIA size from 50 × 70 μm to 350 × 500 μm improved the elongation limit from 135 to 260%. As a comparison, in a previous report, the elongation limit of a single metal layer system was 320%[32]. This is interesting since the shape of the meander was identical to the one reported here. Clearly, the reported VIA limits the stretchability at the current state. Moreover, the location of the VIA within the system has an effect. For example, VIAs between metal tracks show a different size dependent failure rate mechanism (Fig. 3a, c). Again, small (50 × 70 μm) VIAs failed first.

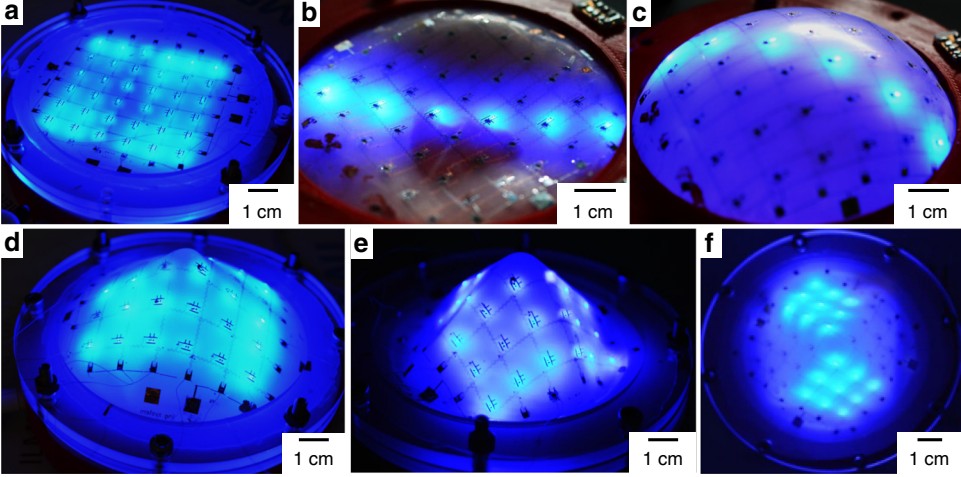

**Fig. 4** Deformable LED active matrix realized using SPCB method. The planar **a** addressable LED matrix can be deformed to different shapes without altering its electrical functionality. The deformation involves deflation (**b**), inflation (**c**), 3D guided shapes (**d**), and vacuum forming (**e**). The addressability of the array remains unaltered after mechanical deformation (**f**)

These results are supported by computer aided stress profile simulations at the VIA locations for different VIA dimensions and conditions (Fig. 3b, d). As can be seen, the local stresses in the metal tracks connected to the VIAs vary gradually. At the same time, the stress concentrations for smaller VIAs are higher than the larger VIAs, which provides higher mechanical stability to the larger VIAs (see Supplementary Fig. 11). Also technically, relatively large VIAs are beneficial since this will result uniform deposition of materials during the electrodeposition process. However, the effects of different locations of VIAs are not well understood from the simulated stress profile. This is probably due to the additional support from the mounted component, which is not the case for the VIAs between two metal tracks.

Figure 3e, f present photographs of an array of the active matrix while the device is in unstretched condition and while being stretched to 220% to its original length using a stress adaptive meander shape metal track, respectively. The design parameters of the metal track are presented in the schematic image (Fig. 3g). Experimentally, we observed that the primary mechanism involved to fail the device is due to the detachment of the metal tracks from the stretchable substrate which eventually results electrical shorts between the metal tracks as shown in Fig. 3h.

It is worth to mention that the rigid components are embedded and the second metal tracks are submerged within the elastomeric matrix, meaning the second metallization tracks are surrounded by the stretchable substrate from three sides. However, the first metallization tracks are only attached with the substrate from one side via the photo-patternable PI layer (see Fig. 2e). Experimentally, we observed that this metal layer is at risk to be peeled off from the elastomeric substrate after prolonged and improper use (Fig. 3h, Supplementary Fig. 12). Full encapsulation using an additional 20 μm thick spin coated layer of EcoFlex decreases the risk of detachment.

The depicted active matrix contains 36 pixels, 6 arrays, and 6 columns of LEDs and transistors within a rubber substrate. In the current array, the pitch of each pixel is 1 cm. However, the spatial resolution of the matrix is not limited by the manufacturing method since the approach uses standard microfabrication techniques to define the metal tracks and the VIAs. The spatial resolution is primarily limited by the physical dimensions of the surface mount components. However, since the approach uses non-stretchable SMDs as functional elements connected via meander-shape metal tracks, the total stretchability of the device depends on the stretchable interconnects and the available stretchable areas in the device, which means that highly dense rigid SMDs will reduce the total stretchability of the system.

On the other hand, direct use of commercial chips is beneficial since those chips are highly developed, integrated and miniaturized with a wide range of functionalities, which eliminates the requirements of developing new electrical components. The miniaturized commercial components in the stretchable electronics will also advance the sensing and actuation capabilities. Furthermore, their integration into free form metamorphic systems will allow to target new applications in robotics, wearable health care, bioengineering, and in biomedicines.

The structure is robust enough to demonstrate deformation behavior. In the device shown, the integrated LED matrix is deformed using different methods to morph from a planar shape (Fig. 4a) to a concave (Fig. 4b) or convex (Fig. 4c) shapes through air deflation or inflation, respectively. Additional deformation involves 3D guided shape or vacuum forming to form cone (Fig. 4d) or to a pyramid (Fig. 4e), respectively (see Supplementary Fig. 13). Since the guiding structure is printed using a 3D printer, a large variety of 3D shapes can be made; other topologies have been demonstrated previously[12,30]. Figure 4f (and Supplementary Fig. 14) shows the addressability of the active LED arrays, where two sides of the pyramid are addressed. In other words, 18 addressing lines, 36 VIAs, and 252 electrical connections to the 72 devices remained intact.

## Discussion
In summary, we reported a design and the fabrication of a stretchable electrical wiring with crossovers and VIAs to isolated devices within an elastomeric matrix. The process shares some similarities with current PCB based fabrication concepts. The goal was to find a solution to transform these commonly rigid structures into a stretchable and deformable counterpart. Currently, only two metallization layers were required from a signal routing point of view. However, the described method can, in principle, be scaled to greater numbers of metal layers with local VIAs in between; a simple solution to prepare "global VIAs", i.e., VIAs

crossing the entire stack is yet to be found. A systematic study led to optimized dimensions of VIA that could achieve a 260% system level elongation. Still the VIA remains the most fragile element from a system level failure point of view.

The recent research trends in stretchable electronics clearly indicate that this emerging technology will not be limited to only lab-based prototypes, it will pursue enormous attraction in commercial products as well. Devices which are already demonstrated in stretchable electronics proof that the technology will find many new types of applications and will improve the performance of many existing devices in various manners. Thus, the limitations remain to be technology related. Stretchable electronics continues to be limited when it comes to a large number of interconnects or multilayer designs with highly integrated electronics which cannot be manufactured reliably for long-term performance using the current state-of-the-art. However, once fully developed, most electronic system known to mankind could be stretchable and could morph to take on new interesting form factors in the future. Many interesting shape adaptive functions could be demonstrated.

## Methods

**Fabrication of multilayer integrated stretchable printed circuit boards.** A 525 μm thick Si wafer (MicroChemicals, Ulm, Germany) was spin coated with a 1 μm thick layer of PMMA (AR-P 6510, Allresist, Strausberg, Germany) and with an 8 μm thick layer of PI (PI 2611, HD Microsystem, Neu-Isenburg, Germany) and cured in a convection oven at 250 °C for 5 hours under $N_2$ flow. 50 nm/200 nm thick layers of Al/Cu was sputter deposited on top of plasma-activated PI layer. The wafer was then patterned for electroplating by photolithography using a negative resist (AZ 15NXT, MicroChemicals, Ulm, Germany). A 10 μm thick layer of Cu was electroplated using Cu 100 electrolyte (NB Technologies, Bremen, Germany). A current density of ~15 mA/cm$^2$ was applied to grow a smooth Cu layer at room temperature. Unwanted Cu and Al was chemically removed using standard Cu and Al etchant (MicroChemicals, Ulm, Germany).

A 20 μm thick photo-patternable polyimide (HD 4100, HD Microsystem, Neu-Isenburg, Germany) was spin coated, baked at 150 °C for 10 min on a hotplate, and patterned for VIA openings by photolithography. A subsequent descumming process (5 min, 50 W, 50 SCCM $O_2$) was performed to remove any residues from the patterned photo-patternable PI in the opening. Then, a 20 μm tall VIA of Cu was electroplated at the openings using the first metal tracks as the seed layer.

Another 20 nm/200 nm of Ti/Cu was sputter deposited as a second metallization layer and the wafer was then patterned for electroplating by photolithography. Another 10 μm thick layer of Cu was electroplated on top of the Cu seed layer. A chemical etching process was carried out to etch the unwanted Ti/Cu. The wafer was then patterned for soldering by photolithography using AZ 1518 positive resist and baked at 120 °C for 10 min. The pads were coated by dip coating in a solder bath (Indalloy #117, MP. 47 °C, Indium Corp., NY).

**Assembly of the SMD components.** The SMD components were assembled on the wafer following a standard pick-and-place technique. The wafer was heated from the back to melt the solder and then the surface mounted components were assembled. Device tests were performed on the wafer to check the interconnections and the device performance was compared before and after the detachment process.

**Encapsulation and detachment.** The silicone EcoFlex (Smooth-On, EcoFlex 00-30) resin was prepared by mixing Part A and Part B (1:1 volume ratio) and by degassing the mixture in a desiccator. The liquid resin of silicone was poured on top of the substrate with assembled components and cured overnight at room temperature. First, EcoFlex was removed manually from the edges of the wafer as EcoFlex is strongly bonded to bare Si. Then the EcoFlex layer was peeled using the sacrificial PI layer. As a final step, the sacrificial PI peeling layer was etched in ECR (40 SCCM $O_2$ + 10 SCCM $CF_4$, 100 W RF power, 0.025 mbar) for 30 min.

## Code availability

The article does not contain any codes. However, the simulation parameters that were used to support the findings of this study are available from the corresponding author upon reasonable request.

## Data availability

The data that support the findings of this study are available from the corresponding author upon reasonable request. The source data for all figures are provided with the paper.

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

## Acknowledgements

The research received financial support through grants from German Science Foundation (JA 1023/3-1, JA1023/8-1, STA556/8-1). S. Biswas would like to thank J. Uziel, I Marquardt, D. Schäfer and B. Hartmann for their help.

## Author contributions

S.B. and H.O.J. conceived the idea, S.B. and A.S. designed and performed the experiments. J.R. and T.S. assisted during the experiments. Y.H. performed the computer simulations. S. B. and J.P. analyzed the data. All authors contributed writing the manuscript.

## Competing interests

The authors declare no competing interests.
