## [Peer Review File · Nature Communications]

Reviewers' comments:

Reviewer #1 (Remarks to the Author):

This manuscript introduces a fabrication scheme to construct stretchable printed circuit boards with multiplexed surface mount devices. The process enables multilayer metal traces with VIAs, and allows the integration of micro transistors with commercial chips to construct active matrix. The technique introduced here is important and useful, but most of the manuscript describes only the fabrication details which are similar with the methods section and are less impactful in terms of scientific merit. I recommend the authors to perform further characterization of their active matrix and demonstrate more sophisticated applications using commercial surface mount devices with industrial-level performances. Here are specific questions and comments that I recommend the authors to address:

1. The main text includes too many engineering details that are similar with the methods section. The authors need to simplify those descriptions throughout the paper and demonstrate more appealing applications of their stretchable active matrix with commercial surface mount devices, and provide more characterizations on the mechanical and electrical properties.
2. The authors use "on-hard-carrier" fabrication to build stretchable printed circuit board with surface mount devices. The same method has been demonstrated before by the same group. What are the differences between the approach and application used in this manuscript and the authors' previous papers?
3. The authors demonstrate a deformable LED active matrix as a proof of concept. However, this demonstration fails to capture the advantages of the authors' platform since there are already a lot of works on stretchable LED array. It will be helpful if the authors can demonstrate an active matrix with more advanced functions using other commercial chips, or demonstrate that the commercial LED array has much better performance compared with other lab-based prototypes.
4. Can the authors increase the spatial resolution of the active matrix based on their approach? What limits the spatial resolution of the authors' active matrix?
5. What are the benefits of using stretchable printed circuit board with commercial chips? Will such systems provide more advanced capabilities in bio-integrated electronics or biomedical related applications? Can the authors demonstrate scenarios that can leverage the stretchability of the platform as well as the advanced sensing/actuation capabilities of the commercial chips?
6. Page 2, line 13, it's not appropriate to state that stretchable electronics are mostly limited to a single active layer. There are already multilayer stretchable electronics integrated transistors demonstrated by different groups. Examples include DOI: 10.1038/NMAT4624, DOI: 10.1038/nature25494, DOI: 10.1038/s41928-018-0116-y.
7. Page 9, line 8, can the authors provide more details on the O₂ plasma activation step? What causes the improved bond strength? Is it due to the -OH group after O₂ plasma treatment or the increased surface roughness? Is it possible to detach the PI layer from PMMA without the O₂ plasma treatment?
8. In Fig. 3, the authors demonstrate that large VIAs result to higher stretchability. What is the mechanics behind this? It is helpful to provide more analysis by either analytical or FEA results. In addition, please include the optical images of the stretchable connections under different levels of elongation, with the description of failure modes at large elongations where necessary.

Reviewer #2 (Remarks to the Author):

This paper presents a fabrication approach for stretchable electronic systems that feature multi-layer metallization. Here, the electronic system consisting of "wavy" interconnects is fabricated through conventional semi-conductor processing techniques on a hard substrate with polymeric insulation layers between multiple layers of metallization. The fabricated circuit is then encapsulated by a stretchable elastomer yielding an overall stretchable circuit. Using this

approach, an addressable LED array is fabricated and tested for electrical and mechanical performance. It was shown that up to over 250% deformation is achieved while maintaining functionality. The via points were identified as the limiting factor for these systems and it was demonstrated that by varying the via size and configuration, mechanical performance of the device can be improved.

This is a well-written paper achieving an intriguing end result of multi-layer stretchable circuits. However, the overall advancement demonstrated in regard to industrially feasible stretchable electronics is rather incremental for this journal. Accordingly, I don't find this paper suitable for Nature Communications.

The primary issue with metal-elastomer composite systems is the drastic mechanical mismatch between the conductors and the encapsulation. This issue particularly affects the multi-cycle deformation of these systems. Authors should be commended for their transparency in stating that delamination between these two elements happen after long term use which demonstrates this point. Even though the electrical functionality of vias and rigid insulation layers are successfully demonstrated here, it is likely that they will add to the aforementioned mechanical issue for multi-cycle deformation.

Another important issue with one of the main premises of the work, which is elastomer encapsulation after complete fabrication of the multi-layer circuit is that for large number of layers and smaller ICs and interconnect spacing, the elastomer precursor will need to fill in very tight spaces. It is known that high compliance elastomers including long polymer chains has precursors with high viscosities that would require very high local pressures to fill in such gaps. A fundamental study on the limitations of the method in this regard would strengthen the work.

Reviewer #3 (Remarks to the Author):

The main claim of the authors is to have developed for the first time a technology for stretchable circuits with multiple conductor layers, allowing circuit track cross overs and VIA connections between the two metal levels. The technology is based on sputtered thin-film layers, which are subsequently electroplated to yield thicker conductors. Components are assembled by soldering technology.

However similar multilayer, thin-film based stretchable circuits technologies have been reported before :

H. Ohmae et al., "Stretchable 45 x 80 RGB LED Display Using Meander Wiring Technology", Proc. Int. Symposium of the Soc. Information Display, San Jose, California, Vol. 46, Issue 1, pp. 102-105, May 31 - June 5, 2015.

R. Verplancke et al., "Stretchable Passive Matrix LED Display with Thin-Film Based Interconnects", Proc. Display Week 2016, San Francisco, paper 49-2, 5 pages, May 22 - 27, 2016.

In these papers up to 4 metal layers have been used. In the same manner as presented in the current paper the circuit is first produced on a temporary carrier and the components are assembled, while the final embedding polymer carrier is applied at the end of the production process. Are the authors aware of this previous work?

However there are also differences with this previously reported work; noteworthy in this respect are

- The use of Si + PMMA as temporary carrier
- The use of electroplating to produce metal tracks with higher conductivity

This indeed yields an original technology, for which the process flow leaves me with a couple of questions :

- The authors obviously try to keep processing temperatures low : (1) the curing of PI2611 is done at 200degC, while the supplier recommended curing temperature is 350degC (2) for component assembly the low melting point (47degC), (expensive) indium (19.1%) and (banned for most applications) lead (22.6%) containing Indalloy #117 solder is used, instead of the (cheaper and lead-free, environmentally friendly) standard SnAgCu (melting temperature 220degC). Why are these choices in materials and process conditions made?
- Is the second plating step (plating of the via) really necessary? This plating step requires connecting all rows (or columns) of the LED matrix in order to do the VIA plating (not that easy to do I would expect). I would expect that direct thin-film deposition of the second metal layer directly on the first metal by sputtering + electroplating would also yield continuous interconnection. Did the authors consider this option?
- Etching the polyimide to transform the flexible circuit into a stretchable one leaves metal 1 tracks supported by polyimide, which will yield stretchable interconnects with high mechanical reliability. However metal 2 tracks are not supported by PI, only by Ecoflex. I therefore expect that the reliability under repeated stretching of metal 2 tracks will be limited. Have the authors tested this ? A possibility to increase the reliability could be to deposit a third PI layer after metal 2. Have the authors considered this option?

Jan Vanfleteren

Reviewer #1 (Remarks to the Author):

This manuscript introduces a fabrication scheme to construct stretchable printed circuit boards with multiplexed surface mount devices. The process enables multilayer metal traces with VIAs, and allows the integration of micro transistors with commercial chips to construct active matrix. The technique introduced here is important and useful, but most of the manuscript describes only the fabrication details which are similar with the methods section and are less impactful in terms of scientific merit. I recommend the authors to perform further characterization of their active matrix and demonstrate more sophisticated applications using commercial surface mount devices with industrial-level performances. Here are specific questions and comments that I recommend the authors to address:

1. The main text includes too many engineering details that are similar with the methods section. The authors need to simplify those descriptions throughout the paper and demonstrate more appealing applications of their stretchable active matrix with commercial surface mount devices, and provide more characterizations on the mechanical and electrical properties.

***Response:** We agree. We have edited the manuscript, shorten the technical details and added more discussions and characterizations of the stretchable devices.*

2. The authors use “on-hard-carrier” fabrication to build stretchable printed circuit board with surface mount devices. The same method has been demonstrated before by the same group. What are the differences between the approach and application used in this manuscript and the authors’ previous papers?

***Response:** This is true that the current manuscript share a similar approach compare with the mentioned previous paper. Both approaches use on-hard-carrier fabrication method. However, the previous manuscript demonstrated only a single metallization layer without any addressing system. While the current manuscript overcome those limitations through realization of a multilayer stretchable printed circuit board with VIA by*

demonstrating an addressable active matrix.

We edited the following section in the manuscript:

Recently, we demonstrated a single layer SPCB method that enabled “on-hard-carrier” fabrication using conventional planar microfabrication techniques that delays use of elastomeric substrate to the end³⁰. The reported method enabled high temperature processing, high alignment and registration, and allowed conventional chip assembly methods on a rigid carrier. However, the previously demonstrated methods used only a single active layer without complex routing of the metal tracks, which limited the complexity of the circuit and the device^{12,30,31}. In this article, we engineered a similar method to realize integrated multilayer SPCB and demonstrate an alternative development towards the realization of stretchable electronics with higher integration density capabilities by introducing stable VIAs through interconnection between different metallization layers. The method used in this article is compatible with conventional micro fabrication processes and uses commercially available pristine SMDs.

Specifically, we report a SPCB method which replaces the rigid insulator substrate of conventional PCB with a highly stretchable silicone elastomer (EcoFlex). To realize VIAs in the SPCB, a similar method to the conventional PCB technology is used here. A highly stretchable (elongated up to 260 % of the original length) multilayered integrated SPCB design is discussed. To demonstrate the applicability, a fully addressable LED active matrix has been realized which is fully addressable. The integrated LED display can be deformed to various three-dimensional (3D) geometrical shapes to morph hemisphere, cone, and pyramid.

3. The authors demonstrate a deformable LED active matrix as a proof of concept. However, this demonstration fails to capture the advantages of the authors’ platform since there are already a lot of works on stretchable LED array. It will be helpful if the authors can demonstrate an active matrix with more advanced functions using other commercial chips, or demonstrate that the commercial LED array has much better performance compared with other lab-based prototypes.

Response: We agree, there are several other demonstrations of stretchable LED arrays. However, we believe that conventional fabrication techniques using conventional materials and commercially available SMDs will advance the stretchable electronics to its next level with improved and reliable functionalities. The depicted method is the first of its kind to demonstrate multilayer stretchable printed circuit boards with VIAs. We used LEDs just to demonstrate the functionality in terms of addressable lightning images and the reliability of the technology.

Secondly, we have previously demonstrated deformable microphone arrays showing that the deformation capability improves certain functionalities of the device. However, that demonstrator used a single metallization layer. The current active LED matrix prototype increases to multilayer approach and in principle, the technology is compatible with any other type of surface mount components.

We edited the following section in the manuscript:

Conventional rigid printed circuit boards (PCB) typically consists of more than one metallization layers to route metal tracks to interconnect surface mount devices (SMDs) using well-established manufacturing methods, which is one of the main reasons behind the paramount success of this technology. On the other hand, stretchable electronics mostly remains limited to a single active layer with less complex device integration, which is primarily due to the lack of reliable manufacturing methods. Although, there are a few lab prototypes of stretchable devices demonstrating multilayer electronic systems with different functionalities²²⁻²⁴, the materials and methods used to realize such devices are unconventional and rarely suitable for industrial production. This technological and materials lacking confines the complexity of demonstrated stretchable electronic devices²⁵. For instance, even the simplest functional active matrix requires at least two metallization layers²⁶.

4. Can the authors increase the spatial resolution of the active matrix based on their approach? What limits the spatial resolution of the authors' active matrix?

Response: Since the depicted approach uses standard microfabrication techniques to define the metal tracks and the VIAs, the spatial resolution is primarily limited by the surface mount devices, meaning, the physical dimensions of the components. However, since the approach uses non-stretchable SMDs as functional elements connected via meander-shape metal tracks, the total stretchability of the device depends on the stretchable interconnects and the available stretchable areas in the device. Highly dense rigid SMDs will reduce the total stretchability of the system.

We added these sentences in the manuscript:

The depicted active matrix contains 36 pixels, 6 arrays and 6 columns of LEDs and transistors within a rubber substrate. In the current array, the pitch of each pixel is 1 cm. However, the spatial resolution of the matrix is not limited by the manufacturing method since the approach uses standard microfabrication techniques to define the metal tracks and the VIAs. The spatial resolution is primarily limited by the physical dimensions of the surface mount components. However, since the approach uses non-stretchable SMDs as functional elements connected via meander-shape metal tracks, the total stretchability of the device depends on the stretchable interconnects and the available stretchable areas in the device, which means that highly dense rigid SMDs will reduce the total stretchability of the system.

5. What are the benefits of using stretchable printed circuit board with commercial chips? Will such systems provide more advanced capabilities in bio-integrated electronics or biomedical related applications? Can the authors demonstrate scenarios that can leverage the stretchability of the platform as well as the advanced sensing/actuation capabilities of the commercial chips?

Response: The main advantage provided by stretchable printed circuit boards compared to their rigid competitors is that they can be mounted on free forms and surfaces which changing morphology during operation, which is not possible in the rigid case. There are several benefits of the direct use of commercial chips. These chips have been developed using highly matured technology and have been well studied for years. On the other hand,

these chips can be miniaturized with highly integrated functionalities. Direct uses of these highly integrated chips will eliminate the requirements of developing new electrical components for stretchable systems. This will accelerate the development of new stretchable devices with wider and reliable functionalities in robotics, wearable health care, bioengineering, or biomedicines.

We added these sentences in the manuscript:

On the other hand, direct use of commercial chips is beneficial since those chips are highly developed, integrated and miniaturized with a wide range of functionalities, which eliminates the requirements of developing new electrical components. The miniaturized commercial components in the stretchable electronics will also advance the sensing and actuation capabilities. Furthermore, their integration into free form metamorphic systems will allow to target new applications in robotics, wearable health care, bioengineering, and in biomedicines.

6. Page 2, line 13, it's not appropriate to state that stretchable electronics are mostly limited to a single active layer. There are already multilayer stretchable electronics integrated transistors demonstrated by different groups. Examples include DOI: 10.1038/NMAT4624, DOI: 10.1038/nature25494, DOI: 10.1038/s41928-018-0116-y.

Response: *We have rewritten the paragraph and added these references.*

We edited the following section in the manuscript:

Conventional rigid printed circuit boards (PCB) typically consists of more than one metallization layers to route metal tracks to interconnect surface mount devices (SMDs) using well-established manufacturing methods, which is one of the main reasons behind the paramount success of this technology. On the other hand, stretchable electronics mostly remains limited to a single active layer with less complex device integration, which is primarily due to the lack of reliable manufacturing methods. Although, there are

a few lab prototypes of stretchable devices demonstrating multilayer electronic systems with different functionalities²²⁻²⁴, the materials and methods used to realize such devices are unconventional and rarely suitable for industrial production. This technological and materials lacking confines the complexity of demonstrated stretchable electronic devices²⁵. For instance, even the simplest functional active matrix requires at least two metallization layers²⁶.

7. Page 9, line 8, can the authors provide more details on the O₂ plasma activation step? What causes the improved bond strength? Is it due to the –OH group after O₂ plasma treatment or the increased surface roughness? Is it possible to detach the PI layer from PMMA without the O₂ plasma treatment?

Response: *The plasma activation process is performed after completing the entire fabrication process, assembly of the SMDs, and on-hard-carrier functionality test; prior to the encapsulation of the device with stretchable silicone mold. The process is carried out without any mask, meaning the whole surface is exposed to the plasma which includes the isolation layer, metal 2, solder, and the SMDs. O₂ plasma has different effects on these different surfaces. In general, the plasma cleans the surfaces, increases the surface roughness and activates the polymer surface with increasing –OH bonding. Please, note that the O₂ plasma treatment is performed to increase the adhesion of the active layers with the stretchable silicone mold. The detachment process does not require any additional process and is not influence due to the plasma process.*

We edited the following section in the manuscript:

To detach the device layer from the hard carrier, we use a castable 3 mm thick and thermo-curable (room temperature, 15 hours) layer of EcoFlex (Smooth-On, EcoFlex 00-30) as a stretchable encapsulation layer which is poured evenly over the entire surface of the fabricated device. To increase the bond strength between the active layers and the EcoFlex, a preceding 5 minutes long O₂ plasma activation step is used. The process is carried out without any mask, meaning the whole surface is exposed to the plasma which

includes the isolation layer, metal 2, solder, and the SMDs. In general, the plasma cleans the surfaces, increases the surface roughness and activates the polymer surface with increasing –OH group.

8. In Fig. 3, the authors demonstrate that large VIAs result to higher stretchability. What is the mechanics behind this? It is helpful to provide more analysis by either analytical or FEA results. In addition, please include the optical images of the stretchable connections under different levels of elongation, with the description of failure modes at large elongations where necessary.

***Response:** We agree. We added several images in figure 3 to support our results with computer simulated stress profile of the VIAs and also added new images showing stretching experiments and failure mechanisms of the device.*

We added the following sections in the manuscript:

*The VIAs play a major role in the system with more than one metallization layers, and the key challenges in the field of multilayer stretchable electronics. The performance of the system largely depends on the reliable VIAs. Specifically, the area of the VIAs had to be optimized in order to achieve fully functioning arrays. The results of this optimization are summarized in **Figure 3** with computer simulated stress profile at the VIA locations while stretched. VIAs connecting bottom and top metal track in an open location (**fig. 3a**) and VIAs connecting a metal track to one of the contact pads of a component (**fig. 3b**) can be distinguished. A goal was to establish the maximum level of uniaxial elongation of the system to cause an electrical discontinuity. The measured values of the elongation ranged up to 260% of the original length can be achieved. Considering VIAs to the contact pads, it was found that maximizing the footprint is beneficial. An increase in the VIA size from $50 \times 70 \mu\text{m}$ to $350 \times 500 \mu\text{m}$ improved the elongation limit from 135% to 260%. As a comparison, in a previous report, the elongation limit of a single metal layer system was 320%³³. This is interesting since the shape of the meander was identical to the one reported here. Clearly, the reported VIA limits the stretchability at the current state. Moreover, the location of the VIA within the system has an effect. For example, VIAs*

between metal tracks show a different size dependent failure rate mechanism (**fig. 3b**). Again, small ($50 \times 70 \mu\text{m}$) VIAs failed first.

These results are supported by computer aided stress profile simulations at the VIA locations for different VIA dimensions and conditions (**fig. 3a₂ & 3b₂**). As can be seen, the local stresses in the metal tracks connected to the VIAs vary gradually. At the same time, the stress concentrations for smaller VIAs are higher than the larger VIAs, which provides higher mechanical stability to the larger VIAs (see supplemental image **fig. S11**). Also technically, relatively large VIAs are beneficial since this will result uniform deposition of materials during the electrodeposition process. However, the effects of different locations of VIAs are not well understood from the simulated stress profile. This is probably due to the additional support from the mounted component, which is not the case for the VIAs between two metal tracks.

Figure 3c presents photographs of an array of the active matrix while the device is being stretched to 220% to its original length using a stress adaptive meander shape metal track. The design parameters of the track are presented in the schematic image. Experimentally, we observed that there are mainly two mechanisms involved to fail the device as shown in the **fig. 3d**. The most frequent failure occurs due to the detachment of the metal tracks from the stretchable substrate (left), and the other is due to the failure of the VIAs. (The failure mechanism is not described, for example breakage or crack formation)

It is worth to mention that the rigid components are embedded and the second metal tracks are submerged within the elastomeric matrix, meaning the second metallization tracks are surrounded by the stretchable substrate from three sides. However, the first metallization tracks are only attached with the substrate from one side via the photo-patternable PI layer (see **fig. 2b₂**). Experimentally, we observed that this metal layer is at risk to be peeled off from the elastomeric substrate after prolonged and improper use (**fig. 3d, left** and supplemental **fig. S11**). Full encapsulation using an additional $20 \mu\text{m}$ thick spin coated layer of EcoFlex decreases the risk of detachment.

We also edited the figure 3 and added computer simulated stress profile of the VIAs, images of stretching tests, and filed devices.

Figure 1 VIA designs affecting the maximum level of elongation, with (a) VIAs connecting bottom and top metal tracks in an open location and (b) VIAs connecting a metal track to one of the contact pads of a SMD; photographs (a₁, b₁), computer simulated stress profile in the metal tracks while elongated 150% to its original length, and the dimensions of the VIAs and corresponding maximum level of uniaxial elongation (table) are shown. (c) Presents the results of the stretching test using a stress adaptive meander shaped metal track (insert) achieving a 220% (up to 260%) elongation to its original length (bottom). (d) Shows the images of different failure modes in the SPCB using current design.

*In the supplemental section we added supporting images of the simulations- **Figure S11***

Figure S11 *Stress profile of VIAs at different locations and dimensions. (a) VIAs connecting bottom and top metal track in an open location and (b) VIAs connecting a metal track to one of the contact pads of a component.*

Reviewer #2 (Remarks to the Author):

This paper presents a fabrication approach for stretchable electronic systems that feature multi-layer metallization. Here, the electronic system consisting of “wavy” interconnects is fabricated through conventional semi-conductor processing techniques on a hard substrate with polymeric insulation layers between multiple layers of metallization. The fabricated circuit is then encapsulated by a stretchable elastomer yielding an overall stretchable circuit. Using this approach, an addressable LED array is fabricated and tested for electrical and mechanical performance. It was shown that up to over 250% deformation is achieved while maintaining functionality. The via points were identified as the limiting factor for these systems and it was demonstrated that by varying the via size and configuration, mechanical performance of the device can be improved.

This is a well-written paper achieving an intriguing end result of multi-layer stretchable circuits. However, the overall advancement demonstrated in regard to industrially feasible stretchable electronics is rather incremental for this journal. Accordingly, I don't find this paper suitable for Nature Communications.

The primary issue with metal-elastomer composite systems is the drastic mechanical mismatch between the conductors and the encapsulation. This issue particularly affects the multi-cycle deformation of these systems. Authors should be commended for their transparency in stating that delamination between these two elements happen after long term use which demonstrates this point. Even though the electrical functionality of vias and rigid insulation layers are successfully demonstrated here, it is likely that they will add to the aforementioned mechanical issue for multi-cycle deformation.

***Response:** We agree. The mechanical mismatch between the active elements of the device and the encapsulated materials, which is in this case EcoFlex, is an issue. However, a large scientific community in this field is actively addressing this issue and demonstrated improved performance. An alternative approach of using rigid metal conductors might be liquid alloys, which is also being studied by several research groups. However, each of these approaches has their limitations. One strong argument supporting rigid metals as*

electrodes is that they are conventional materials with very high electrical performance, which is not the case for the liquid alloys.

We have reported different stretchable electronic systems using metals as conductive interconnects demonstrating more than ten thousand stretching and release cycles at 150% elongation level. This could be further improved. The current article reports a similar design with increased active layer and complex electrical routing. However, due to the increased complexity of the device, the multicycle deformation of the current device is significantly less. This issue of the current system is mentioned in the manuscript.

We added these sentences in the main manuscript and added figures showing stretching and failures.

It is worth to mention that the rigid components are embedded and the second metal tracks are submerged within the elastomeric matrix, meaning the second metallization tracks are surrounded by the stretchable substrate from three sides. However, the first metallization tracks are only attached with the substrate from one side via the photo-patternable PI layer (see fig. 2b₂). Experimentally, we observed that this metal layer is at risk to be peeled off from the elastomeric substrate after prolonged and improper use (fig. 3d, left and supplemental fig. S11). Full encapsulation using an additional 20 μm thick spin coated layer of EcoFlex decreases the risk of detachment.

Another important issue with one of the main premises of the work, which is elastomer encapsulation after complete fabrication of the multi-layer circuit is that for large number of layers and smaller ICs and interconnect spacing, the elastomer precursor will need to fill in very tight spaces. It is known that high compliance elastomers including long polymer chains has precursors with high viscosities that would require very high local pressures to fill in such gaps. A fundamental study on the limitations of the method in this regard would strengthen the work.

Response: *We agree. Elastomeric resin with high viscosity will require very high local pressure to fill small gaps. In our current demonstrator, and many other research groups, we used EcoFlex 00-30 (viscosity, $\eta = 3000$ cP). We also tested these process with PDMS ($\eta = 3500$ cP) which shows similar “filling” properties. EcoFlex and PDMS are also well known for their micro-patterning applications through soft-lithography, where the*

silicone resins required to fill micrometer—scale features. We have performed some new experiments to verify the “filling” properties of EcoFlex using micrometer scale gaps and depths. The results are added in the supplemental sections. The EcoFlex 00-30 could “fill” gaps less than 10 μm with a 10 μm depth (see the attached SEM image in the supplemental section). In our demonstrated device this dimensions are more than 50 μm each.

We added these sentences in the main manuscript and added supplemental figures.

As shown schematically in **fig. 2a₁** (and supplemental **fig. S₈**), the molding process encapsulates the SMDs, meaning the EcoFlex under fill the SMDs. It is known that elastomers with high viscosities will require high local pressures to fill in small gaps. However, we observed that EcoFlex 00-30 (viscosity, $\eta = 3000$ cP) can fill smaller gaps than 10 μm (see supplemental **fig. S₉**). Comparing with Polydimethylsiloxane (PDMS), which is well known for micro-patterning through soft-lithography, has a viscosity of 3500 cP. Both of these two polymers are strong candidates as a stretchable substrate for high dense stretchable electronics.

Figure S9 Under-filling of EcoFlex. EcoFlex 00-30 can under-fill very small gaps. (a) A back view of a LED embedded in EcoFlex showing the under-filling of EcoFlex in a gap of approximately 50 μm (schematic) and (b) a SEM image of a test structure showing that EcoFlex under-filled gaps of less than 10 μm (schematic).

Reviewer #3 (Remarks to the Author):

The main claim of the authors is to have developed for the first time a technology for stretchable circuits with multiple conductor layers, allowing circuit track cross overs and VIA connections between the two metal levels. The technology is based on sputtered thin-film layers, which are subsequently electroplated to yield thicker conductors. Components are assembled by soldering technology.

However similar multilayer, thin-film based stretchable circuits technologies have been reported before :

H. Ohmae et al., “Stretchable 45 x 80 RGB LED Display Using Meander Wiring Technology”, Proc. Int. Symposium of the Soc. Information Display, San Jose, California, Vol. 46, Issue 1, pp. 102–105, May 31 – June 5, 2015.

R. Verplancke et al., “Stretchable Passive Matrix LED Display with Thin-Film Based Interconnects”, Proc. Display Week 2016, San Francisco, paper 49-2, 5 pages, May 22 - 27, 2016.

In these papers up to 4 metal layers have been used. In the same manner as presented in the current paper the circuit is first produced on a temporary carrier and the components are assembled, while the final embedding polymer carrier is applied at the end of the production process. Are the authors aware of this previous work?

Response: *We have added new references including this.*

However there are also differences with this previously reported work; noteworthy in this respect are

- The use of Si + PMMA as temporary carrier
- The use of electroplating to produce metal tracks with higher conductivity

This indeed yields an original technology, for which the process flow leaves me with a couple of questions :

- The authors obviously try to keep processing temperatures low : (1) the curing of PI2611 is done at 200degC, while the supplier recommended curing temperature is 350degC (2) for component assembly the low melting point (47degC), (expensive) indium (19.1%) and (banned for most applications) lead (22.6%) containing Indalloy #117 solder is used, instead of the (cheaper and lead-free, environmentally friendly) standard SnAgCu (melting temperature 220degC). Why are these choices in materials and process conditions made?

Response: We agree. We cure the PI2611 at 200 C for 5 hrs under N₂, while the supplier recommends at 350 C. This change in the processing temperature has been selected because in the depicted demonstrator the PI2611 acts only as a sacrificial layer, the final device does not contain any PI2611. The fully cured PI is recommended while the material remains as a part of the device. Also, plasma etching rate varies for fully cure PI and partially cure PI.

Although the manufacturing methods allow high temperature processing, we used a low melting point solder (Indalloy #117). The reason behind this is the method of solder coating. In the depicted approach, we used a parallel dip-coating process in a heated liquid solder bath to coat the contact pads. However, a higher melting point solder could be used as well following other solder printing methods.

We added these sentences in the manuscript:

*The solder bumps are used to make mechanical and electrical contact in between the metal tracks and the active components. A low melting point solder (Indalloy #117) is used for this purpose because the solder is applied by a parallel dip-coating process in a liquid solder bath using a 10 μm thick photoresist mask to define the solder bump locations (see supplemental **fig. S6**). However, a higher melting point solder could be used as well following other solder printing methods.*

- Is the second plating step (plating of the via) really necessary? This plating step requires connecting all rows (or columns) of the LED matrix in order to do the VIA plating (not that easy to do I would expect). I would expect that direct thin-film deposition of the second metal layer

directly on the first metal by sputtering + electroplating would also yield continuous interconnection. Did the authors consider this option?

***Response:** The second electroplating step is necessary to ensure good electrical and stable mechanical contact in between the first metal tracks and the VIAs, which would be challenging to achieve using thin film deposition methods through 20 μm deep holes (VIAs). The approach presented in this manuscript is inspired by conventional PCB technologies aiming to realize a method that would be compatible with industrial manufacturing methods. The electroplated VIAs presented here follows similar approach to the standard PCB technology.*

We edited the following sentences in the manuscript:

*As mentioned earlier, metal tracks in different layers require to be interconnected through VIAs and realizing reliable VIAs in SPCB remained a challenge till today. In the depicted approach, like the conventional PCB method, we use electroplated Cu to realize the VIAs. The photo-patternable PI layer is also photolithographically patterned to define openings and locations of the VIAs, which are subsequently filled with 20 μm of electrodeposited copper. This second electrodeposition step is necessary in order to ensure good electrical contact in between the first metal tracks and the VIAs, which is difficult to achieve using thin-film deposition methods through 20 μm deep holes. A similar electrodeposition method is also used in conventional PCB technologies to grow thick VIAs. As shown schematically in **fig. 1c₁**, the VIAs are grown using metal 1 as a seed layer through predefined openings in the photo-patternable PI layer. A plasma cleaning process is required prior to electrodeposit the VIAs in order to remove the residues from photo-patternable PI (supplemental **fig. S₅**) to ensure good electrical and stable mechanical contact between different metal layers in these regions.*

- Etching the polyimide to transform the flexible circuit into a stretchable one leaves metal 1 tracks supported by polyimide, which will yield stretchable interconnects with high mechanical reliability. However metal 2 tracks are not supported by PI, only by Ecoflex. I therefore expect that the reliability under repeated stretching of metal 2 tracks will be limited. Have the authors

tested this ? A possibility to increase the reliability could be to deposit a third PI layer after metal 2. Have the authors considered this option?

Response: *We agree that metal 2 tracks are not supported by PI, only by EcoFlex. This is also true that the reliability under repeated stretching of metal 2 tracks is limited. Experimentally we observed that the metal 2 peeled off easily from the stretchable polymer which leads to random deformation of the metals resulting electrical short. This is a drawback of the current demonstrator. There are a couple of solutions to this problem. One solution could be, as the reviewer pointed out, to encapsulate the metal 2 with another layer of PI. One of our previous demonstrator followed this approach for single layer SPCB. Another solution could be, as mentioned in the manuscript, to encapsulate the entire system by another layer of EcoFlex.*

However, from our previous experiences we were aware about this problem. To address this issue, we introduced an additional plasma activation step prior to encapsulation of the device layer by stretchable mold. This plasma activation step increases the bonding strength between the metals and the polymer, which results a higher stretching and release cycle.

We edited the following sentences in the manuscript:

It is worth to mention that the rigid components are embedded and the second metal tracks are submerged within the elastomeric matrix, meaning the second metallization tracks are surrounded by the stretchable substrate from three sides. However, the first metallization tracks are only attached with the substrate from one side via the photo-patternable PI layer (see fig. 2b₂). Experimentally, we observed that this metal layer is at risk to be peeled off from the elastomeric substrate after prolonged and improper use (fig. 3d, left and supplemental fig. S₁₂). Full encapsulation using an additional 20 μm thick spin coated layer of EcoFlex decreases the risk of detachment.

REVIEWERS' COMMENTS:

Reviewer #1 (Remarks to the Author):

The authors have addressed all of the comments from the referees. I feel that the revised manuscript is suitable for publication.

Reviewer #2 (Remarks to the Author):

I appreciate the authors' response and the extra work they put into the revision. However, I still don't think the my main concern is addressed. Authors themselves are acknowledging that the increase in complexity, which is the main merit of this paper, leads to a an important problem of lack of mechanical reliability. This unfortunately diminishes the impact of this work below the level that would be appropriate for this journal thus I don't recommend publication. I do believe, however, that this paper would be a good fit for a more specialized journal on device technologies.

Reviewer #3 (Remarks to the Author):

My comments have been properly addressed by the authors, I have no additional remarks.

Jan Vanfleteren

Reviewer #1 (Remarks to the Author):

The authors have addressed all of the comments from the referees. I feel that the revised manuscript is suitable for publication.

***Response:** Thank you for your valuable comments and suggestions.*

Reviewer #2 (Remarks to the Author):

I appreciate the authors' response and the extra work they put into the revision. However, I still don't think the my main concern is addressed. Authors themselves are acknowledging that the increase in complexity, which is the main merit of this paper, leads to a an important problem of lack of mechanical reliability. This unfortunately diminishes the impact of this work below the level that would be appropriate for this journal thus I don't recommend publication. I do believe, however, that this paper would be a good fit for a more specialized journal on device technologies.

***Response:** Thank you. However, we disagree with this concern.*

The reported manuscript demonstrates heterogeneous materials and methods to enable the fabrication of multilayer integrated stretchable printed circuit boards. The method enables the realization of multiple stretchable metallization layers and introduces a solution to prepare vertical interconnect access (VIA) connections between the layers and to the device segments. Considering current state-of-the-art, the demonstrated method is

beneficial compare to the other methods since it enables on-hard carrier fabrication using conventional planar microfabrication techniques which would be compatible for the industrial manufacturing. It is shown that by maximizing the footprint of the via for a certain interconnect design in the reliability of the multilayer stretchable printed circuit board the reliability of the metamorphic system can be improved. The depicted method allows to increase the complexity of the stretchable circuit boards and is a step towards higher integration density. This has been demonstrated through realization of an addressable active matrix with integrated LEDs and transistors within a rubber substrate. The final device is capable of reversible deformations which has been shown through a number of 3D shapes and geometries.

The article reports sufficient novel results and discusses the limitations of the existing systems, and demonstrates their solution. In depth study of the methods and materials to realize stretchable printed circuit boards is reported. The study includes optical and scanning electron microscope (SEM) analysis of the materials, and characterizing their electrical and mechanical properties. The experimental results are also supported by the computer simulated models where necessary.

Since reviewer did not ask any particular question regarding the manuscript, we did not change anything in the manuscript.

Reviewer #3 (Remarks to the Author):

My comments have been properly addressed by the authors, I have no additional remarks.

Response: *Thank you for your valuable comments and suggestions.*